# 2S-BUSGAN: A Novel Generative Adversarial Network for Realistic Breast Ultrasound Image with Corresponding Tumor Contour Based on Small Datasets

**DOI:** 10.3390/s23208614

**Published:** 2023-10-20

**Authors:** Jie Luo, Heqing Zhang, Yan Zhuang, Lin Han, Ke Chen, Zhan Hua, Cheng Li, Jiangli Lin

**Affiliations:** 1College of Biomedical Engineering, Sichuan University, Chengdu 610065, China; roger_luojie1107@163.com (J.L.); gjzzcc@163.com (L.H.); chenke@scu.edu.cn (K.C.); 2Department of Ultrasound, West China Hospital, Sichuan University, Chengdu 610065, China; zhangheqing16@sina.com; 3Highong Intellimage Medical Technology (Tianjin) Co., Ltd., Tianjin 300480, China; 4China-Japan Friendship Hospital, Beijing 100029, China; huazhan@hotmail.com (Z.H.); licheng20080118@sina.com (C.L.)

**Keywords:** breast ultrasound, deep learning, data augmentation, generative adversarial network, image synthesis, small datasets

## Abstract

Deep learning (DL) models in breast ultrasound (BUS) image analysis face challenges with data imbalance and limited atypical tumor samples. Generative Adversarial Networks (GAN) address these challenges by providing efficient data augmentation for small datasets. However, current GAN approaches fail to capture the structural features of BUS and generated images lack structural legitimacy and are unrealistic. Furthermore, generated images require manual annotation for different downstream tasks before they can be used. Therefore, we propose a two-stage GAN framework, 2s-BUSGAN, for generating annotated BUS images. It consists of the Mask Generation Stage (MGS) and the Image Generation Stage (IGS), generating benign and malignant BUS images using corresponding tumor contours. Moreover, we employ a Feature-Matching Loss (FML) to enhance the quality of generated images and utilize a Differential Augmentation Module (DAM) to improve GAN performance on small datasets. We conduct experiments on two datasets, BUSI and Collected. Moreover, results indicate that the quality of generated images is improved compared with traditional GAN methods. Additionally, our generated images underwent evaluation by ultrasound experts, demonstrating the possibility of deceiving doctors. A comparative evaluation showed that our method also outperforms traditional GAN methods when applied to training segmentation and classification models. Our method achieved a classification accuracy of 69% and 85.7% on two datasets, respectively, which is about 3% and 2% higher than that of the traditional augmentation model. The segmentation model trained using the 2s-BUSGAN augmented datasets achieved DICE scores of 75% and 73% on the two datasets, respectively, which were higher than the traditional augmentation methods. Our research tackles imbalanced and limited BUS image data challenges. Our 2s-BUSGAN augmentation method holds potential for enhancing deep learning model performance in the field.

## 1. Introduction

Breast ultrasound (BUS) has been widely used in breast cancer diagnosis [1,2]. In recent years, auxiliary diagnosis research based on Deep Learning (DL) has developed rapidly and played an essential role in breast cancer diagnosis [3,4]. Generally, DL requires large-scale annotated datasets to produce an effective model. However, this is difficult for medical data, especially for BUS data. On the one hand, although a large amount of BUS data exists, the collection of BUS data is complex due to patient privacy, and the labeling of the data is very time-consuming and requires professional physicians [5,6]. On the other hand, BUS data take a long-tailed distribution form, and images of benign masses are much more prevalent than those of malignant or normal masses.

Extensive research shows that data augmentation overcomes challenges posed by small and imbalanced datasets [7]. Two main methods are used: Traditional Augmentation (TA) and Synthetic Augmentation (SA). TA applies common operations such as translation, rotation, brightness adjustment, scaling, and clipping to expand the dataset [8,9,10]. However, TA has limitations in generating entirely new patterns. SA uses existing data to generate new samples, employing techniques such as interpolation-based methods (e.g., SMOTE [11] and Sample Pairing [12]) and generation-based augmentation (e.g., Generative Adversarial Networks [13]). SA incorporates additional information that is not present in the original data, overcoming TA’s limitations. Figure 1 depicts results from a liver tumor classification study by Frid Adar M et al. [10] on their CT dataset. While TA and additional real data show modest accuracy improvements (red and green curves), SA (blue curve) significantly enhances accuracy.

As a powerful generation-based augmentation method, GAN has played a remarkable role in medical data distribution. In recent medical research, several GAN variants, including DCGAN [14], LSGAN [15], WGAN [16], and others, have been developed and applied to diverse tasks such as classification [17], segmentation [3], reconstruction [18], and synthesis [19]. These methods have demonstrated their effectiveness in augmenting breast ultrasound (BUS) datasets and improving the performance of deep learning models, particularly in classification and segmentation tasks.

For instance, Al Dhabyani et al. [19] employed GAN for BUS dataset augmentation and showed superior performance compared with traditional augmentation methods when used with CNN and transfer learning. However, the limited size of the training data can lead to the generation of similar results, limiting the effectiveness of dataset augmentation. One practical solution for the small dataset problem is augmenting the dataset with unlabeled data. Pang et al. [9] proposed a semi-supervised deep learning model based on GAN, utilizing unlabeled BUS images for classification. They demonstrated that this approach could generate high-quality BUS images, significantly enrich the dataset, and achieve more accurate breast cancer classification results. Sudipan Saha and Nasrullah Sheikh et al. [20] tackled the small dataset challenge by proposing a BUS classification model using the Auxiliary Classifier Generative Adversarial Network (ACGAN) [21]. They trained the GAN model by augmenting the BUS dataset using traditional augmentation methods, improving classification performance.

In a different approach, Ruixuan Zhang et al. [18] utilized partially masked BUS images to generate new images, employing an image data augmentation strategy based on mask reconstruction. This method effectively increased the interpretability and diversity of disease data while maintaining structural integrity. However, it is important to note that the augmented datasets using these augmentation methods may exhibit significant differences from the original data distribution, which could impact downstream tasks.

The performance of most breast ultrasound (BUS) segmentation methods heavily relies on accurately labeled data, which can be challenging to obtain. As a solution, many segmentation methods leverage unlabeled data to expand the dataset and utilize GAN-based approaches to enhance model performance.

For instance, Luyi Han et al. [17] proposed a semi-supervised deep-learning segmentation model for BUS images. They input the unlabeled BUS image data into a GAN to generate corresponding labeled data. Similarly, Donghai Zhai et al. [3] employed an asymmetric generative adversarial network (ASSGAN) comprising two generators and a discriminator. This architecture enables mutual supervision between the generators and generates reliable segmentation-predicted masks as guidance for unlabeled images. Comparative studies demonstrated that these methods effectively improve segmentation performance even with a limited number of labeled images.

However, it is worth noting that these semi-supervised methods require a substantial amount of unlabeled data. Additionally, if a classification task is involved, both augmentation methods cannot be reused and require relabeling.

In summary, there are still some challenges to the generation-based augmentation method applied to BUS image data:Image quality: BUS images exhibit diverse and random morphological features of tumors and texture features of surrounding tissues, leading to unrealistic details and a lack of structural legitimacy in the generated images.Data constraints: The limited size of the BUS dataset affects both the performance of deep learning models and GANs. Since GANs rely on the underlying data distribution, the small dataset needs to accurately represent the true data distribution, resulting in insufficient diversity and limited information in the generated results.Application limitations: The current GAN augmentation methods have a significant constraint when applied to different tasks. They are typically tailored for specific tasks, such as classification or segmentation. However, if these augmented data need to be utilized for other tasks, the data must be relabeled, or a new GAN model must be trained, leading to increased costs. Consequently, a universal generation-based augmentation method is meaningful.

In response to these challenges, our research objectives can be summarized as follows: We aim to address the small imbalance in ultrasound breast tumor data by expanding the dataset through data generation. Additionally, we seek to enhance the robustness and generalization of ultrasound breast tumor image recognition and segmentation.

Therefore, we propose a two-stage GAN framework to address the above challenges, called 2s-BUSGAN. The framework consists of the Mask Generation Stage (MGS) and the Image Generation Stage (IGS), which enables the generation of realistic BUS images and their corresponding tumor contour images, eliminating the need for relabeling. To overcome the data constraints, we employ a Differential Augmentation strategy (DiffAug) [22], which enhances the generalization of GANs for small datasets. Additionally, we incorporate a feature-matching loss that minimizes the discrepancy between real and generated images at a high-level feature representation, thereby improving the quality of synthesized images.

Our 2s-BUSGAN model departs from common GAN models by adopting a two-stage framework, consisting of MGM and IGM. This approach provides a straightforward means of distinguishing between surrounding tissues and tumor regions, enhancing the interpretability of the generated images. Moreover, our generated results include breast tumor images and corresponding tumor contour images. This novel work avoids re-annotating tumor regions in generated images, and can effectively enhance the performance of models that require tumor region annotations for training, such as segmentation models.

## 2. Materials and Methods

In this section, we present the 2s-BUSGAN model. The goal was to generate realistic images with tumor contour annotations and structural legitimacy. The model was trained on imbalanced and small BUS datasets. Using a single GAN model to achieve our goal would make the model training unstable. Therefore, we divided the model into two stages: Mask Generation Stage (MGS) and Image Generation Stage (IGS), implementing the mapping process from random noises to tumor contours to the corresponding BUS images. 

### 2.1. Overview of 2s-BUSGAN

The flowchart of our method is shown in Figure 2. Our proposed 2s-BUSGAN consists of four components: Mask Generation Stage (MGS), Image Generation Stage (IGS), Differentiable Augmentation Module (DAM), and Feature Matching Loss (FML). First, MGS uses random noises to generate realistic pseudo-tumor contour images. Second, IGS generates BUS images based on real and generated tumor contour images. Then, FML enriches the detail information of the synthesized images, and DAM enhances the generalization of 2s-BUSGAN. The following sections will describe these components. 

### 2.2. Mask Generation Stage (MGS)

The MGS is the initial phase of 2s-BUSGAN and generates synthetic tumor contour images from random noise input. Achieving stable GAN training has been an ongoing challenge in GAN model research, and even the simplest models can be challenging to train stably. Therefore, in the design and selection of the MGM model, we conducted extensive experiments, including the choice of different models, parameter adjustments, the selection of various GAN loss functions, the use of different normalization layers, and the selection of activation functions. The MGS is a simplified noise-to-image synthesis GAN comprising a generator *G*_1_ and a discriminator *D*_1_. The architectural configuration is depicted in Figure 3a. The generator provides a random noise following a normal distribution as input, while the output is a pseudo tumor contour image. Similar to other general generative models, generator *G*_1_ architecture encompasses five up-sampling blocks, each comprising a deconvolution layer with a stride of 2, a batch normalization layer (BN) [23], and a ReLU [24] activation layer, as illustrated in the left section of Figure 3a. All deconvolution layers employ a kernel size of 4 × 4, and the number of kernels progressively decreases from 512 to 32. Additionally, the output layer employs the hyperbolic tanh activation function, ensuring values are bounded from −1 to 1.

In discriminator *D*_1_, we incorporate spectral normalization (SN) layers into its architecture. Spectral normalization is an effective technique to enhance training stability in GANs by normalizing the parameters of each convolution layer, ensuring the GAN satisfies the 1-Lipschitz condition [25]. Spectral normalization is a principled, easy-to-implement approach that has been successfully used in many excellent models. Therefore, the structure of *D*_1_ consists of six down-sampling blocks, each composed of a convolution layer, a spectral normalization layer, a LeakyReLU [26] activation function, and a dropout layer, as depicted in the right section of Figure 3a. The convolutional kernel size is set to 4 × 4, and the filters progressively decrease from 512 to 1. As the first stage of 2s-BUSGAN, the loss of MGS will be optimized gradually:(1)$$L_{G_{1}}=E_{z\sim P(z)}[\log\left(1-D_{1}(G_{1}(z))\right)]$$
(2)$$L_{D_{1}}=E_{{y\sim P}_{data}(y)}[\log D_{1}(y)]+E_{z\sim p(z)}[\log\left(1-D_{1}(G_{1}(z))\right)]$$
where *L_G_*_1_ and *L_D_*_1_, respectively, represent the losses of generator *G*_1_ and discriminator *D*_1_, *z* denotes the random vectors, *y* denotes the tumor contour images of ground truth BUS images, and *G*_1_*(z)* denotes the synthesized tumor contour images from input *z*. The objective function can represent the similarity of the synthesized image in quality to the real image.

### 2.3. Image Generation Stage (IGS)

After the MGS, the generated tumor contour image is a binarized matrix, where 0 represents the missing region, and 1 represents the background. However, it solely captures the morphological and structural information of the tumor, lacking texture details. Hence, the Image Generation Stage (IGS) is introduced to generate realistic BUS images by incorporating texture information. This stage addresses the image reconstruction problem, aiming to produce complete BUS images based on the provided structural information. 

Similar to MGS, IGS comprises a generator *G*_2_ and a discriminator *D*_2_. In contrast to other generative models that utilize noise as input, this module aims to achieve image-to-image generation using tumor contour images as input. Generator *G*_2_ employs an encoder-decoder architecture with n residual blocks [27]. This architecture consists of four down-sampling blocks and four up-sampling blocks, as shown in the left part of Figure 3b. Each block incorporates a convolution or deconvolution layer with a stride of two, an instance normalization layer (IN) [28], and a ReLU activation function. The architecture includes 9 residual blocks, with each block consisting of two convolution layers with a stride of 1. The first and last convolution layers adapt the channel size using a 1 × 1 kernel size, while all other convolution and deconvolution layers use a 3 × 3 kernel size. For discriminator *D*_2_, each input is a concatenation of the tumor contour image and its corresponding BUS image. To assess image quality at a local scale, we utilize a discriminator with a specific structure inspired by PatchGAN [29]. Unlike general discriminators that output a single scalar for global judgment, this discriminator produces a matrix that evaluates image quality across multiple scales. Each unit in the output matrix represents the likelihood that an image patch is authentic, capturing local information. This design allows for the incorporation of high-frequency details into the generator. 

Consequently, the discriminator structure comprises four down-sampling blocks, as depicted in the right part of Figure 3b. Each block includes a convolution layer, spectral normalization (SN) layer, LeakyReLU activation function, and dropout layer. Notably, we apply SN to each convolution layer of the discriminator to enhance its performance, similar to discriminator *D*_1_ of MGS. The convolution kernel size is set to 4 × 4, and the number of filters decreases from 512 to 1 sequentially.

The loss function can be expressed as:(3)$$L_{G_{2}}=E_{y\sim P(y)}[\log\left(1-D_{2}(G_{2}(y))\right)]$$
(4)$$L_{D_{2}}=E_{{x,y\sim P}_{data}(x,y)}[\log D_{2}(x,y)]+E_{G_{2}(y),y\sim p(G_{2}(y),y)}[\log\left(1-D_{2}(G_{2}(y),y)\right)]$$
where *L_G_*_2_ and *L_D_*_2_, respectively, represent the losses of generator *G*_2_ and discriminator *D*_2_, *y* denotes tumor contour images of ground truth BUS images, and *G*_2_*(y)* denotes the synthesized BUS images from tumor contour images *y*. 

### 2.4. Feature Matching Loss (FML)

IGS faces challenges in learning the accurate mapping of tumor regions in BUS images due to the inherent randomness in the location and size of tumors. Moreover, the generated images exhibited blurriness and noise, particularly in the regions surrounding the tumors, compared to the ground truth images. To address these issues and enhance image synthesis quality, we introduced a feature-matching loss that imposes additional constraints based on the discriminator of IGS.

The feature-matching loss plays a crucial role in stabilizing the training process and significantly improving the performance of GAN. It is closely related to perceptual loss [30,31], which has found widespread application in style transfer research and image super-resolution [32,33,34]. This loss function enforces similarity between the generated BUS images and real BUS images across multiple scales, ensuring that the generator generates images that accurately capture the key characteristics of the dataset. To extract features from the BUS images at different scales, we utilize the discriminator of IGS as a feature extractor, employing its multi-layer architecture.

Thus, the feature-matching loss can be described as follows:(5)$$L_{FM}(G_{2},D_{2})=E_{{x,y\sim P}_{data}(x,y)}\sum_{i=1}^{M}\frac{1}{N_{i}}[{\left\|D_{2}^{\left(i\right)}(x,y)-D_{2}^{\left(i\right)}(G_{2}(y),y)\right\|}_{1}]$$
where *M* is the total number of layers, and *N_i_* is the number of elements in each layer, and $D_{2}^{\left(i\right)}$ is discriminator *D*_2_’s *i*th-layer feature extractor. 

Full objective combines both the loss of generator *G*_2_ and feature matching loss are defined as follows: (6)$$\underset{G}{\min}\left(L_{G_{2}}+\lambda L_{FM}\right)$$
where *λ* is a constant of the loss of the feature matching loss, and its values are set to 10.0 in our study. 

### 2.5. Differentiable Augmentation Module (DAM)

The performance of GANs depends on sufficient and well-distributed training data. Lack of training data can lead to overfitting the GAN and limited diversity in the generated samples. Traditional augmentation (TA) methods, which modify the dataset using non-differentiable operations, alter the data distribution and can negatively affect the performance of GANs trained on such augmented data.

To overcome this limitation, we employ Differentiable Augmentation (DiffAug) [22], which allows for the simultaneous augmentation of real and generated image data using invertible and differentiable operations. By leveraging differentiable operations, DA ensures that the network can be fully trained without altering the distribution of the original data. Therefore, the DAM incorporates DiffAug and updates the loss functions as follows:(7)$$L_{G_{1}}=E_{z\sim P(z)}[\log\left(1-D_{1}(T(G_{1}(z)))\right)]$$
(8)$$L_{D_{1}}=E_{{y\sim P}_{data}(y)}[\log D_{1}(T(y))]+E_{z\sim p(z)}[\log\left(1-D_{1}(T(G_{1}(z)))\right)]$$
(9)$$L_{G_{2}}=E_{y\sim P(y)}[\log\left(1-D_{2}(T(G_{2}(y)))\right)]$$
(10)$$L_{D_{2}}=E_{{x,y\sim P}_{data}(x,y)}[\log D_{2}(T(x,y))]\;+E_{G_{2}(y),y\sim p(G_{2}(y),y)}[\log\left(1-D_{2}(T(G_{2}(y),y))\right)]$$
where *T* denotes differentiable augmentation operations, such as brightness adjustment and translation. Noteworthy, the generated and real samples are subjected to the same differentiable augmentation operation.

## 3. Experiments

In this section, we conducted qualitative and quantitative evaluations. First, we conducted experiments of the image generation performance of MGS and IGS within the 2s-BUSGAN framework. Moreover, we evaluated 2s-BUSGAN’s overall generative capacity. Second, we submitted the generated images for expert medical evaluations, focusing on image quality and the discernment of benign and malignant features. Lastly, we employed the images produced by 2s-BUSGAN as augmented data to gauge their influence on model performance in breast malignancy classification and breast segmentation.

### 3.1. Datasets

In our study, we use two different and representative datasets of breast ultrasound (BUS) images: the BUSI [19,35] dataset and the Collected dataset (obtained from a de-identified source at a hospital). Both datasets contain BUS images and corresponding segmentation masks annotated by experienced medical professionals. The dataset has undergone data cleaning to ensure the removal of any patient information. Figure 4 provides examples of the images and their corresponding segmentation maps. The details of each dataset are described as follows:

In each category of images, on the left are the real US images, and on the right are the annotated tumor contour images.

BUSI [19,35]. This dataset was curated by the National Cancer Center of Cairo University in Egypt in 2018. The dataset consists of 780 PNG images obtained using the LOGIQ E9 and LOGIQ E9 agile ultrasound systems. These images were acquired from 600 female patients aged between 25 and 75. Among the images, 487 represent benign tumors, 210 represent malignant tumors, and 133 contain normal breast tissue. The labels for benign and malignant tumors were determined based on pathological examination results from biopsy samples. The average image size is 500 × 500 pixels, and each case includes the original ultrasound image data and the corresponding breast tumor boundary image annotated by an expert imaging diagnostic physician. The annotated tumor contour images serve as the gold standard for segmentation in the experiments, providing a reference for the training process and evaluation of results.Collected. The China-Japan Friendship Hospital in Beijing collected and organized the dataset in 2018. Before data collection, informed written consent was obtained from each patient after a comprehensive explanation of the study’s procedures, purpose, and nature. Multiple acquisition systems were used to capture the images in this dataset. It comprises a total of 920 images, including 470 benign cases and 450 malignant cases. The benign or malignant classification labels are derived from pathological examinations based on puncture biopsy results. A professional imaging diagnostic physician has annotated each data case. Furthermore, the dataset includes notable misdiagnoses and missed diagnosis cases, providing valuable examples for analysis and evaluation.

### 3.2. Implementation Details

Our experimental setup utilized the PyTorch framework on a single NVIDIA GeForce RTX 2080 Ti GPU with 10 G GPU memory. We employed a batch size of 16 and utilized the Adam optimizer [36]. Generator *G*_2_ and discriminator *D*_2_ had learning rates of 0.0001 and 0.0004, respectively. The weight *λ* for the feature-matching loss term in generator *G*_2_, the number of residual blocks in *G*_2_, and the output size of *D*_2_ was set to 10, 5, and 16 × 16 for the BUSI dataset, and 10, 5, and 16 × 16 for the Collected dataset. For the BUSI dataset, generator *G*_2_ utilized a weight *λ* of 10 for the FML term, included 5 residual blocks, and output size of *D*_2_ was 16 × 16. Similarly, for the Collected dataset, the same settings were employed with a weight *λ* of 10 for the FML term, 5 residual blocks, and output size of *D*_2_ was 16 × 16. 

In our study, we set the weight *λ* to 10. This value was chosen with reference to related literature on feature matching loss. In practical experiments, we compared four different values: 0.1, 1, 10, and 100. Through experimentation, we found that setting it to 10 produced better results. As a result, we chose to use the referenced weight *λ*.

Due to limitations with GPU, we were unable to train models when the number of residual blocks exceeded 5. We compared the performance of models with different numbers of residual blocks, ranging from 1 to 5. Our experiments revealed that the best results were achieved with 5 residual blocks; thus, we decided to set this parameter to 5. 

All images in both datasets were resized to 256 × 256 pixels. In general, larger images contain more information, resizing images can significantly affect a model’s learning and performance. Unfortunately, our experimental setup was constrained by GPU memory capacity. Given these constraints, images resizing to 256 × 256 pixels are the most suitable compromise. This adjustment not only optimized GPU memory utilization but also streamlined downstream model processing for various tasks. 

The models were trained for 6000 epochs across all stages. To ensure distinctive class features in the generated images, we separately trained the benign and malignant BUS images for all datasets.

Furthermore, our experiments involved the use of classification and segmentation models, as well as comparative generative models. For these models, we adhered to the hyperparameter settings found in their respective reference literature to ensure experimental correctness. Therefore, no further hyperparameter tuning was conducted for these models.

We comprehensively evaluated our proposed method using two distinct BUS datasets. The evaluation encompassed both quantitative and qualitative analyses of the experimental results. For the qualitative assessment, we conducted a visual comparison between the real BUS images and the synthesized images. We utilized three numerical metrics: Fréchet Inception Distance (FID) [37], Kernel Inception Distance (KID) [38], and multi-scale structural similarity (MS-SSIM) [39] to quantitatively evaluate the performance. FID and KID are metrics used to measure the similarity between images based on their features. While both metrics assess feature-level distance, KID estimates are unbiased. Lower values of FID and KID indicate better performance in terms of image similarity. MS-SSIM is a metric that quantifies the similarity between paired images on a scale from 0 to 1. Higher MS-SSIM values indicate a greater degree of similarity between the images. In our experiments, we calculated the internal MS-SSIM of the generated data to gauge the diversity of the generated samples [40]. Lower MS-SSIM values indicate richer diversity in the generated data. The evaluation metrics are described as follows:(11)$$FID(x,g)={\left\|\mu_{x}-\mu_{g}\right\|}_{2}^{2}+Tr(\sum_{x}+\sum_{g}-2\sqrt{(\sum_{x}\sum_{g})})$$
(12)$$KID(x,g)=\frac{1}{m(m-1)}\sum_{i\neq j}^{m}k(v_{x_{i}},v_{x_{j}})+\frac{1}{n(n-1)}\sum_{i\neq j}^{n}k(v_{g_{i}},v_{g_{j}})\;-\frac{1}{mn}\sum_{i=1}^{m}\sum_{i\neq j}^{n}k(v_{x_{i}},v_{g_{j}})$$
(13)$$MS-SSIM(x,g)={\left[l_{M}(x,g)\right]}^{\alpha_{M}}\cdot \prod_{j=1}^{M}{\left[c_{j}(x,g)\right]}^{\beta_{j}}\cdot {\left[s_{j}(x,g)\right]}^{\gamma_{j}}$$
where *x* denotes real images, *g* denotes generated images, *μ* and Σ denote means and covariances of intermediate layer features of real and generated images, *m* is the sample size of the generated images, *n* is the sample size of the real image, *k* denotes the kernel function, *v* is the feature vector of real and generated images. *M* is the size of the intermediate layer of feature extractor, *l*, *c*, and *s* represent the brightness, contrast, and structure between images, respectively, and *α*, *β*, and *γ* represent weights.

### 3.3. Generation Experiments

#### 3.3.1. MGS Mask Synthesis

MGS effectively maps random noise samples from a normal distribution to the distribution of tumor contour images. In Figure 5, we compare real tumor contour images and the generated images produced by MGS, including both benign and malignant data. Both methods were trained for an equal number of epochs: 3000.

In addition, we utilized the t-SNE (t-distributed stochastic neighbor embedding) [41] algorithm to visualize the disparities between synthetic and real data, as illustrated in Figure 6. The t-SNE algorithm is a nonlinear dimensionality reduction technique capable of visualizing high-dimensional data into two or three dimensions.

#### 3.3.2. IGS Image Synthesis

The generator of IGS exhibits the capability to capture the texture feature distribution within the BUS image by inpainting both the internal and external regions of the tumor contour. By reconstructing real BUS images from real masks, IGS demonstrates its effectiveness. Additionally, we compared real BUS images and the results reconstructed from real masks by IGS and IGS without differential augmentation before the discriminator (IGS w/o DAM).

The main task of IGS is to convert the breast tumor contour images generated by MGS into realistic BUS images. Therefore, we compare generated from the random mask by IGS, IGS without the feature-matching loss (IGS w/o FML), and IGS without differential augmentation before the discriminator (IGS w/o DAM). The number of training epochs was set to 6000 for both the Collected and BUSI datasets.

#### 3.3.3. 2s-BUSGAN Image Synthesis

In this experiment, we utilized 2s-BUSGAN, incorporating MGS and IGS, to generate BUS images from random noise and compare their performance with conventional generative augmentation methods. Furthermore, we employed the t-SNE method to compare the distribution differences between the original data and the data generated by various generation methods.

### 3.4. Evaluation by Doctors

To assess the quality of the synthesized BUS images, we conducted a subjective evaluation by randomly selecting 120 images of different types. These images comprised 60 ground truth (GT) images and 60 generated images from different methods, including LSGAN (the most effective traditional generative augmentation method mentioned in Section 3.5) and 2s-BUSGAN, with each type consisting of 15 benign and 15 malignant images. Three experienced ultrasonographers conducted an independent evaluation and a benign and malignant classification of these images.

### 3.5. Augmentation Experiments for BUS Segmentation and Classification

The original dataset was divided into training, validation, and test subsets for segmentation and classification experiments, with a ratio of 60%, 20%, and 20%, respectively. Due to the imbalanced distribution of benign and malignant cases in the BUSI dataset, we utilized the oversampling method [7] to balance the dataset. Three-fold cross-validation was employed for training the classification models to evaluate their performance. Online augmentation [42] was applied to both segmentation and classification experiments, with a probability of 0.3 for each augmentation during the training process. In the classification experiment, augmentation techniques such as translation, horizontal flipping, and rotation were utilized, which preserved the morphological characteristics of the tumor. For the segmentation experiment, augmentation operations included rotation, blurring, translation, gamma transformation, scaling, blurring, scaling, and adding Gaussian noise.

#### 3.5.1. Classification Experiments

We compared different methods based on various augmentation techniques: Baseline (no augmentation), traditional augmentation (TA), traditional augmentation combined with LSGAN-based augmentation (TA + LSGAN), and traditional augmentation combined with 2s-BUSGAN (TA + 2s-BUSGAN). Our classification model of choice was VGG16 [43], which had been pre-trained on the ImageNet dataset and fine-tuned using only real data. During training, we kept the parameters of the first three layers of VGG16 frozen and only trained the last and fully connected layers. For the evaluation of classification performance in the presence of class imbalance, we utilized precision, recall, F1 score, and accuracy as the evaluation metrics, and they are described as follows: (14)$$Precision=\frac{TP}{TP+FP}$$
(15)$$Recall=\frac{TP}{TP+FN}$$
(16)$$Accuracy=\frac{TP+TN}{TP+FN+TN+FP}$$
(17)$$F1-Score=2\cdot \frac{Precision\cdot Recall}{Precision+Recall}$$
where TN denotes the number of true negative samples.

#### 3.5.2. Segmentation Experiments

We conducted experiments to assess the impact of different augmentation methods on the segmentation performance. Three settings were compared: no augmentation (Baseline), traditional augmentation (TA), and TA plus proposed 2s-BUSGAN (TA + 2s-BUSGAN). The segmentation model used U-Net [44], and the performance was evaluated using the Dice, which is described as follows.
(18)$$Dice=\frac{2\cdot TP}{2\cdot TP+FP+FN}$$
where TP denotes the number of true positive samples, FN denotes the number of false negative samples, and FP denotes the number of false positive samples.

## 4. Results

### 4.1. Generation Experiments Results

#### 4.1.1. MGS Mask Synthesis Results

From Figure 5, it is evident that there is a discernible distinction between benign and malignant tumor contour images. Additionally, MGS generates images that visually resemble real tumor contour images.

In Figure 6, points are the generated masks (red points) and real masks (blue points) after the dimensionality reduction. These generated data points occupy positions between the original data points, and are close to them. Moreover, this data distribution phenomenon occurs in both BUSI datasets and Collected datasets.

#### 4.1.2. IGS Image Synthesis Results

Figure 7 compares real BUS images and the results reconstructed from real masks by IGS and IGS without differential augmentation before the discriminator (IGS w/o DAM). The IGS w/o DAM exhibits minimal differences from the real images. The generated results by IGS exhibit slight differences in their details compared with the real images. Furthermore, both IGS w/o DAM and IGS can reconstruct the tumor region and the surrounding tissue based on the real mask. Comparing the results of tumor region reconstruction between these two methods, the reconstruction by IGS shows more differences from the real tumor area than the reconstruction by IGS w/o DAM. This phenomenon also applies to the reconstruction of surrounding tissue, with a pronounced example seen in reconstructed malignant images from the Collected dataset of the (c) and (d) columns. In (c), the reconstruction effectively captures areas with no information in the real images (top-left and top-right corners), while (d) exhibits random tissue texture reconstruction.

Figure 8 illustrates the generation results of different GAN models using various datasets. Upon comparing the methods, we observe that IGS w/o FML generates the lowest-quality images, confined to localized regions of the tumor contour. Moreover, IGS w/o DAM generates images of higher quality than IGS w/o FML, albeit still exhibiting blurred tumor surrounding tissues and increased noise. Compared to the generation results of other methods, images generated by IGS exhibit the highest quality in terms of generation. While ensuring the quality of tumor region generation, the surrounding tissue area also demonstrates more structural legitimacy in its texture. Moreover, the tissue texture closely resembles that of real images, with minimal image noise.

The quantitative results of different models of IGS are shown in Table 1. IGS w/o FML exhibits the poorest performance across all metrics for all datasets. Additionally, IGS w/o DAM performs better than IGS w/o FML across all metrics but remains inferior to IGS. Notably, IGS attains the most favorable results across all metrics.

#### 4.1.3. 2s-BUSGAN Image Synthesis Results

Figure 9 showcases real BUS images alongside images generated by 2s-BUSGAN, DCGAN, LSGAN, and WGAN. As depicted in Figure 9, most exhibit improved generation quality among the conventional methods. However, they still exhibit specific issues, such as blurring of the tumor’s surrounding tissues and the tumor area and the significant similarity between the generated images. In Figure 8, the images generated by WGAN appear more blurred and of lower quality. Conversely, the generated images by 2s-BUSGAN will be more realistic.

Table 2 presents the quantitative results of different GAN models. In the quantitative results on the Collected dataset, LSGAN outperforms other conventional methods in terms of both the FID and KID metrics but lags in the MS-SSIM metric. On the BUSI dataset, LSGAN excels in FID and KID metrics compared with other conventional methods only when comparing the generation of malignant images, while DCGAN leads in various metrics over other conventional methods when comparing the generation of benign images. 2s-BUSGAN closely approximates the performance of the best conventional method across all evaluation metrics on different datasets. Moreover, in the MS-SSIM metric, 2s-BUSGAN consistently outperforms other conventional methods.

As illustrated in Figure 10, benign images (red points) and malignant images (blue points) after the dimensionality reduction are very close to each other in both the BUSI dataset and the Collected dataset. In contrast, most generated data points from different generation methods exhibit clear separation from the distribution of image data generated by these conventional GAN methods. The data distribution of 2s-BUSGAN reveals a partial overlap between the distributions of benign and malignant data, and is closer to the real data distribution.

### 4.2. Results of Evaluation by Doctors

The results of the subjective evaluation conducted by the three doctors are presented in Table 3. The accuracy represents the percentage of correct classifications for each image type, real or fake. A lower value indicates greater difficulty in distinguishing the authenticity of images. The accuracy of the GT images is generally higher, as the doctors are more familiar with real BUS images. However, the accuracy of GT images from the Collected dataset will be lower than that of the BUSI dataset.

In comparing the accuracy of the generated images from different generation methods, we found that the accuracy of LSGAN and 2s-BUSGAN from the Collected dataset was lower than those from the BUSI dataset. In addition, we found that the accuracy of generated images of 2s-BUSGAN is higher than that of LSGAN.

As depicted in Table 4, there is no significant variation in the outcomes across the doctors. Comparing the results within the same dataset, we observe higher accuracy for the BUSI dataset than the Collected dataset. When comparing the different methods, we find that the LSGAN method achieves high classification accuracy in the BUSI dataset, with Doctor 1 achieving a 100% correct rate. In contrast, the accuracy of images generated by LSGAN and 2s-BUSGAN methods in the Collected dataset is similar. However, the 2s-BUSGAN method demonstrates closer alignment with the classification accuracy of the real images.

### 4.3. Results of Augmentation Experiments for BUS Segmentation and Classification

#### 4.3.1. Classification Experiments Results

Table 5 presents the performance metrics of classification models trained on augmented datasets using different augmentation methods. Classification models trained on the Collected dataset performed less than those trained on the BUSI dataset. Although the overall performance did not significantly improve with traditional augmentation and GAN methods, there was notable improvement when employing the 2s-BUSGAN method.

#### 4.3.2. Segmentation Experiments Results

Figure 11 provides visualizations of the segmentation results obtained using different augmentation methods. When training the segmentation model on a small dataset alone (baseline), the resulting segmentation differs significantly from the ground truth (GT) mask. However, when augmentation methods augment the small dataset, the segmentation model’s performance improves noticeably. The segmentation results obtained using the 2s-BUSGAN method closely resemble the GT mask. 

Table 6 presents the segmentation results of BUS images using the baseline approach and various augmentation methods. Across all datasets, applying different data augmentation methods leads to improved performance in the segmentation experiments. However, traditional augmentation (TA) only brings about marginal performance improvements in the segmentation model. On the other hand, significant enhancements are observed when employing the 2s-BUSGAN method. 

In conclusion, we observed the efficacy of both MGS and IGS in our image generation experiments. MGS successfully generates tumor contour images closely resembling real ones, and the data distribution of generated masks aligns well with real masks. IGS excels at reconstructing BUS images, even without differential augmentation, and stands out as the top performer in image quality among various GAN models. Moreover, our doctor evaluations validate the clinical relevance of 2s-BUSGAN, with high accuracy in line with real images. This underscores its potential practical use in the medical field. In our augmentation experiments for BUS segmentation and classification, 2s-BUSGAN outshines traditional methods, substantially enhancing model performance and segmentation results.

## 5. Discussion

### 5.1. Generation Experiments Discussion

#### 5.1.1. MGS Mask Synthesis Discussion

The notable distinction observed in benign and malignant tumor contour images, as revealed in Figure 5, suggests that our GAN models can distinguish between benign and malignant data. These generated images bear a striking resemblance to real tumor contour images.

The results from Figure 6 showcase the striking resemblance between generated masks and their real counterparts after dimensionality reduction. The distribution of the generated and original data points signifies the effectiveness of the MGS. This finding indicates that the generated masks exhibit a similar distribution to the real masks, and the generated data can effectively augment the original data. Thus, MGS can generate images that are similar to real masks and have similar distributions.

#### 5.1.2. IGS Image Synthesis Discussion

The results presented in Figure 7 highlight the potential of IGS to effectively reconstruct real breast ultrasound images from real breast tumor contour images. This is a significant finding indicating that our approach can generate images with a high degree of realism, with implications for improving image data quality and enhancing the performance of deep learning models.

The reconstruction of detailed information poses a challenge due to the inherent randomness in the position and size of tumor regions in breast tumor boundary images. The IGS w/o DAM exhibits minimal differences from the real images, indicating that IGS possesses inherent capabilities for effective image reconstruction. However, the DiffAug component in DAM introduces a randomness to the model during training, resulting in generated results by IGS with DAM exhibiting slight differences in the details compared with the real images. While this may not be ideal for image reconstruction, it serves the purpose of effectively augmenting the data.

We can observe the importance of the FML component in Figure 8. The absence of FML in IGS hampers the generation of meaningful textures in areas distant from the tumor contour. FML plays a crucial role in ensuring that the generated image closely resembles the real image across multiple scales, not just in the vicinity of the tumor area. FML helps ensure that generated images closely resemble real images across multiple scales, not limited to the tumor area. This is crucial for generating highly realistic images.

In conclusion, IGS not only reconstructs breast ultrasound images from real breast tumor contour images but also generates realistic BUS images from generated breast tumor contour images. We conducted ablation experiments on IGS to compare the effect of removing the key components, FML and DAM. The reconstruction effect of IGS is lower than that of the method of removing DAM because the introduction of the DAM module increases the randomness of the images generated by IGS. When FML is removed, the quality of image generation is much lower than other methods. The results of quantitative and qualitative experiments proved that these two components are effective.

#### 5.1.3. 2s-BUSGAN Image Synthesis Discussion

As depicted in Figure 9, most exhibit improved generation quality among the conventional methods. However, they still exhibit specific issues, such as blurring of the tumor’s surrounding tissues and the tumor area and the significant similarity between the generated images. In Figure 8, the images generated by WGAN appear more blurred and of lower quality. Conversely, the generated images by 2s-BUSGAN will be more realistic.

Results of Table 2 demonstrate that 2s-BUSGAN can generate BUS images of comparable quality to traditional methods while ensuring adequate diversity. Additionally, 2s-BUSGAN has the advantage of generating corresponding breast tumor contour images alongside the BUS images, making the results more comprehensive and practical.

In Figure 10, it is hard to distinguish benign and malignant points in both the BUSI dataset and the Collected dataset. This is due to inherent feature similarities between benign and malignant BUS images in the original dataset, making accurate classification difficult. In contrast, generated data points from different traditional GAN methods have good separation, indicating distinct differences in benign and malignant data features of generated images that are easily discernible. However, the data distribution of 2s-BUSGAN is closer to the original data distribution than other generation methods, indicating 2s-BUSGAN better represent the real data distribution.

In conclusion, the results of quantitative and qualitative experiments demonstrate that the quality of 2s-BUSGAN generated images is similar to that of the generated images of the traditional GAN methods, and outperforms the other methods in some metrics. In addition, comparing the distribution of the amplified data of various methods, it can be found that the 2s-BUSGAN augmented dataset is closer to the real dataset. Therefore, 2s-BUSGAN not only generates high-quality BUS images, but also generates tumor contour annotations at the same time, which has a wider scope of application.

### 5.2. Discussion of Evaluation by Doctors

The BUS images generated by 2s-BUSGAN have the potential to deceive experienced doctors regarding authenticity and benign versus malignant features. The subjective evaluation involving doctors highlights that generated images from both LSGAN and 2s-BUSGAN are closely to real images where experienced medical professionals may find it challenging to distinguish between the two. There are many manually introduced artifacts, such as rectangular borders, cross labels, and blood flow labels, making it difficult for the model to generate these markers correctly and meaningfully. The inclusion of artifacts and variances in image quality between datasets influences the doctors’ judgment. The performance of 2s-BUSGAN in this context is marginally superior to LSGAN, indicating its effectiveness in producing highly convincing synthetic BUS images.

### 5.3. Discussion of Augmentation Experiments for BUS Segmentation and Classification

#### 5.3.1. Classification Experiments Discussion

The results from Table 5 highlight some key findings in the context of classification models trained on augmented datasets. Notably, the performance of classification models was superior when using the BUSI dataset as compared to the Collected dataset. This discrepancy could be attributed to variations in data quality, labeling, or other dataset-specific factors. Moreover, while traditional augmentation and GAN methods provided some improvement in classification model performance, the most significant enhancement was observed when employing the 2s-BUSGAN method. This result underscores the effectiveness of the proposed 2s-BUSGAN framework in augmenting the BUS dataset, resulting in improved classification accuracy for distinguishing benign and malignant cases.

Traditional augmentation techniques and basic GAN methods have limitations in effectively enhancing classification model performance. In contrast, the 2s-BUSGAN approach addresses these limitations and presents a promising solution for improving the classification accuracy of BUS images. 

#### 5.3.2. Segmentation Experiments Discussion

The visual demonstration in Figure 11 intuitively illustrates that using data augmentation methods, particularly image synthesis methods, effectively enhances the performance of segmentation models trained with limited datasets. The segmentation results presented in Table 6 similarly corroborates this finding.

Our proposed two-stage generation approach, which generates tumor contour images and subsequently transforms them into BUS images from random vectors, effectively enhances the segmentation model’s performance. Moreover, our method eliminates the need for additional labeling work on the generated BUS images, thus expanding its applicability to a broader range of scenarios.

Extensive qualitative and quantitative experiments conducted on the BUSI and Collected datasets have proved the effectiveness of our method. However, there are still areas for optimization in our method. One aspect is the network complexity of 2s-BUSGAN, which prolongs the overall training time. Additionally, as the training of MGS is independent of IGS, the potential information cannot be effectively propagated back to MGS, potentially leading to biases between the generated data and the original data distribution. Consequently, optimizing the model and fully extracting the information in BUS data represent urgent future research directions.

In conclusion, our GAN framework, consisting of MGS, IGS, and 2s-BUSGAN, excels in generating breast ultrasound and tumor contour images. IGS can effectively reconstruct images, and the inclusion of FML and DAM components enhances image quality. 2s-BUSGAN narrows the gap between benign and malignant images, yielding realistic and clinically relevant results. The doctor evaluation affirms the convincing quality of images produced by 2s-BUSGAN, making it challenging for experts to distinguish from real images. It outperforms conventional GAN methods. Our augmentation experiments showcase the limitations of traditional methods and the significant improvements offered by 2s-BUSGAN, enhancing classification accuracy, especially in distinguishing benign from malignant cases. In segmentation, 2s-BUSGAN effectively boosts model performance and eliminates the need for extra labeling work, broadening its application potential.

Moreover, there are still areas for optimization in our method. One aspect is the network complexity of 2s-BUSGAN, which prolongs the overall training time. Additionally, as the training of MGS is independent of IGS, the potential information cannot be effectively propagated back to MGS, potentially leading to biases between the generated data and the original data distribution. Consequently, optimizing the model and fully extracting the information in BUS data represent urgent future research directions.

## 6. Conclusions

In this research paper, we addressed challenges of imbalanced and insufficient BUS image data by proposing a two-stage generative augmentation method called 2s-BUSGAN. Our 2s-BUSGAN method consists of two stages: MGS and IGS. These stages enable the generation of tumor contour images and subsequent transformation into BUS images from random noise inputs. Additionally, we incorporated FML to enhance the performance of 2s-BUSGAN and enrich the background texture information of the generated images. Moreover, DAM was employed to enhance the generalization ability of 2s-BUSGAN, particularly in scenarios with small sample datasets.

Through extensive qualitative and quantitative experiments conducted on the BUSI and Collected datasets, we demonstrated the effectiveness of 2s-BUSGAN. The results confirmed that our proposed method is a versatile augmentation technique that significantly improves the performance of deep learning models, including segmentation and classification while circumventing the need for data relabeling. 

Overall, our study contributes to addressing the challenges associated with imbalanced and limited BUS image data, and our proposed 2s-BUSGAN method shows promise in augmenting BUS image datasets and enhancing the performance of deep learning models in the field.

## Figures and Tables

**Figure 1 sensors-23-08614-f001:**
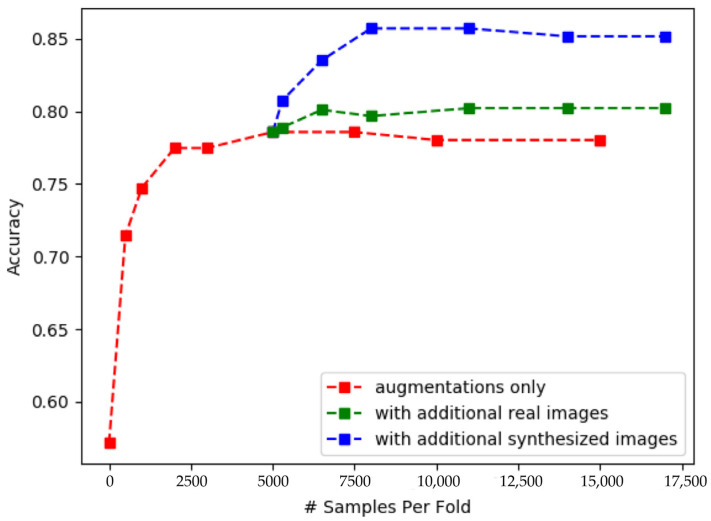
Total accuracy results for liver lesion classification. The x-axis represents the size of the training set, and the y-axis indicates classification accuracy. The red line shows the effect after adding traditional data augmentation samples, the green line shows the effect after adding a small amount of real data, and the blue line shows the effect after adding synthetic data based on GAN. (Reprinted with permission from Ref. [10]. Copyright 2018, Elsevier B.V.)

**Figure 2 sensors-23-08614-f002:**
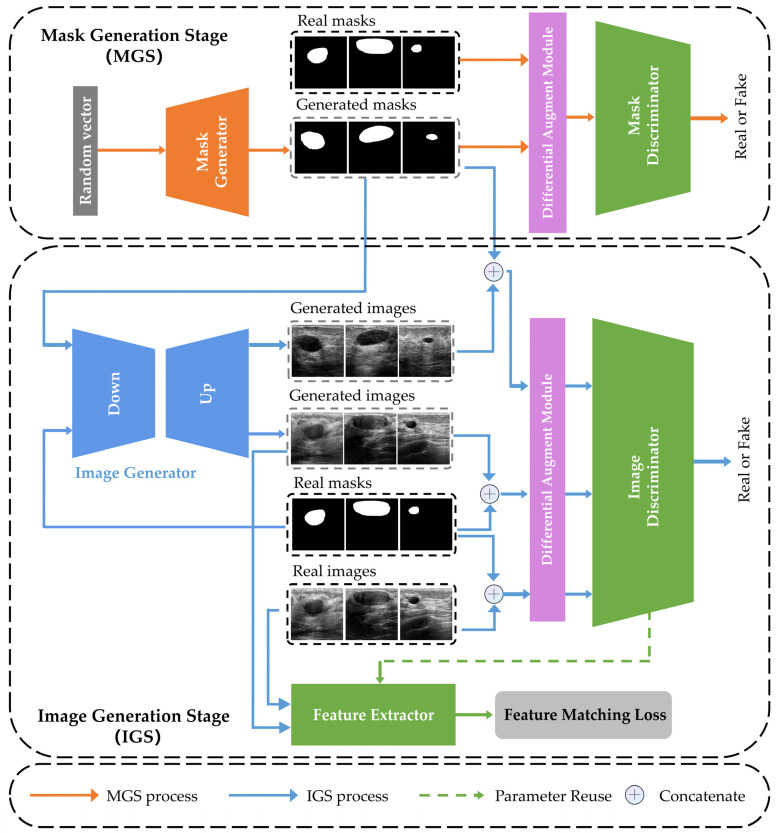
The framework of our proposed 2s-BUSGAN.

**Figure 3 sensors-23-08614-f003:**
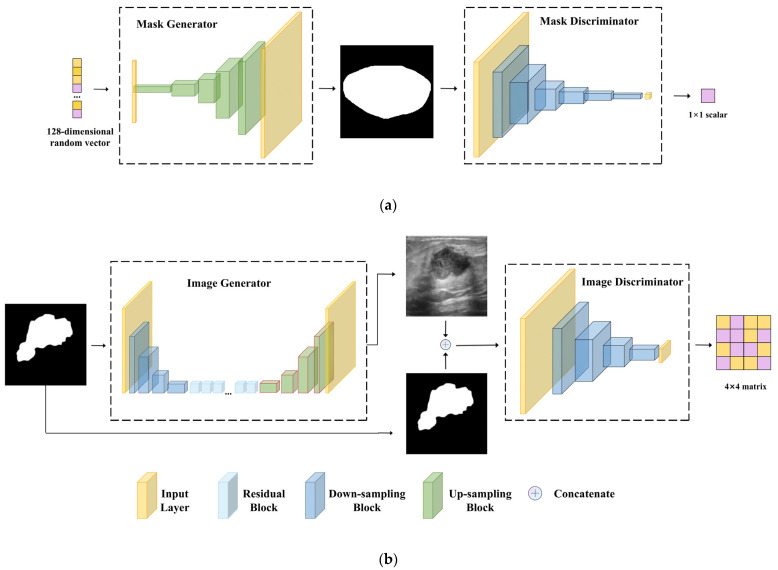
Architectures of (**a**) MGS and (**b**) IGS.

**Figure 4 sensors-23-08614-f004:**
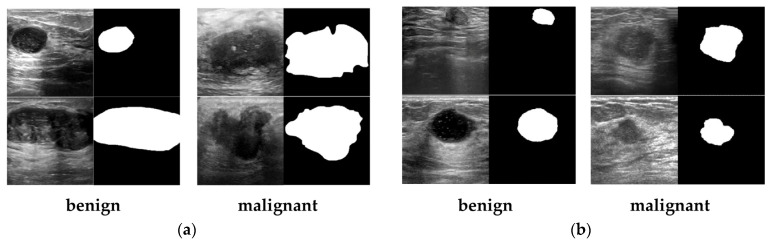
Examples of the BUS images and the corresponding tumor contour images of different categories: (**a**) BUSI dataset and (**b**) Collected dataset.

**Figure 5 sensors-23-08614-f005:**
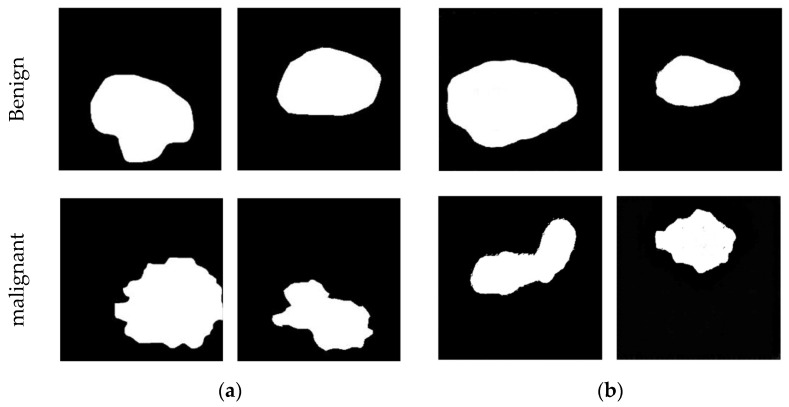
Examples of mask synthesis: (**a**) examples of real masks and (**b**) examples generated by MGS.

**Figure 6 sensors-23-08614-f006:**
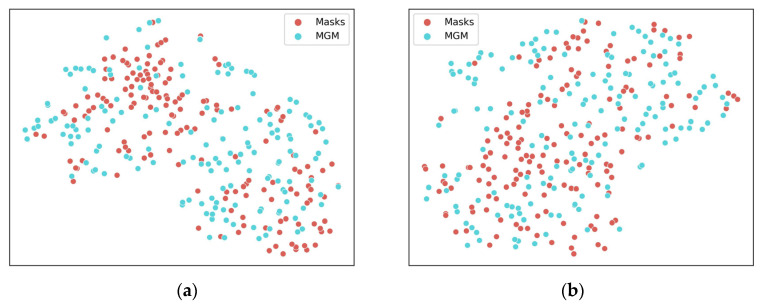
T-SNE visualization of synthetic and real masks for two datasets: (**a**) BUSI dataset and (**b**) Collected dataset.

**Figure 7 sensors-23-08614-f007:**
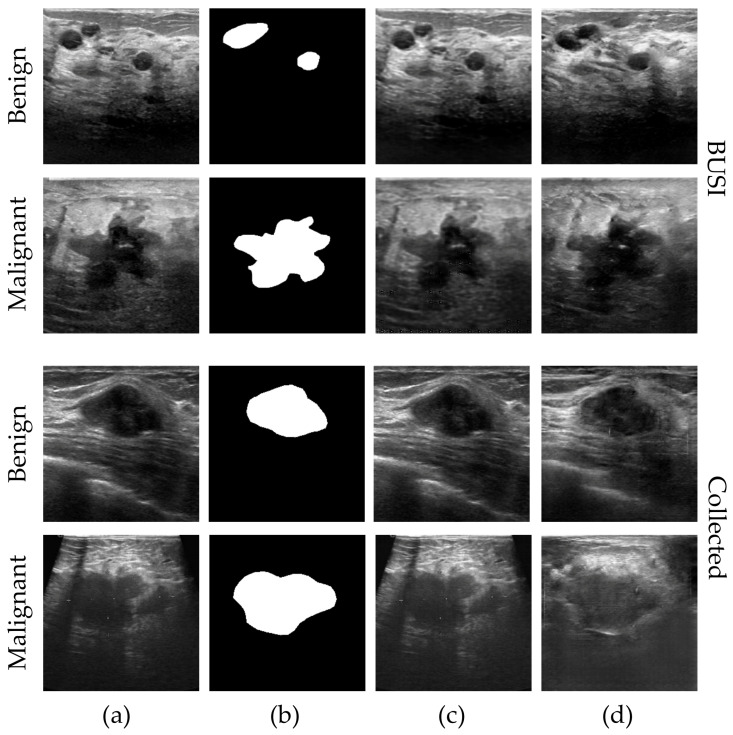
Examples of BUS image reconstruction from real masks for two datasets. From left to right, they are GT images and masks and generated results for different methods: (**a**) real images, (**b**) real masks, (**c**) images generated by IGS w/o DAM, and (**d**) images generated by IGS.

**Figure 8 sensors-23-08614-f008:**
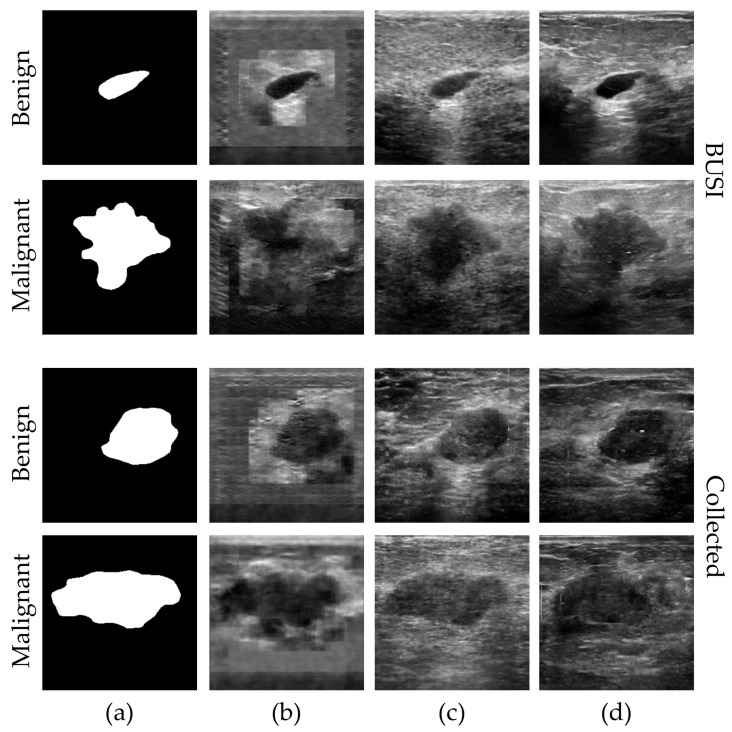
Examples of BUS image synthesis from generated masks for two datasets. From left to right, they are: (**a**) GT masks and generated results of different methods, (**b**) images generated by IGS w/o FML, (**c**) images generated by IGS w/o DAM, and (**d**) images generated by IGS.

**Figure 9 sensors-23-08614-f009:**
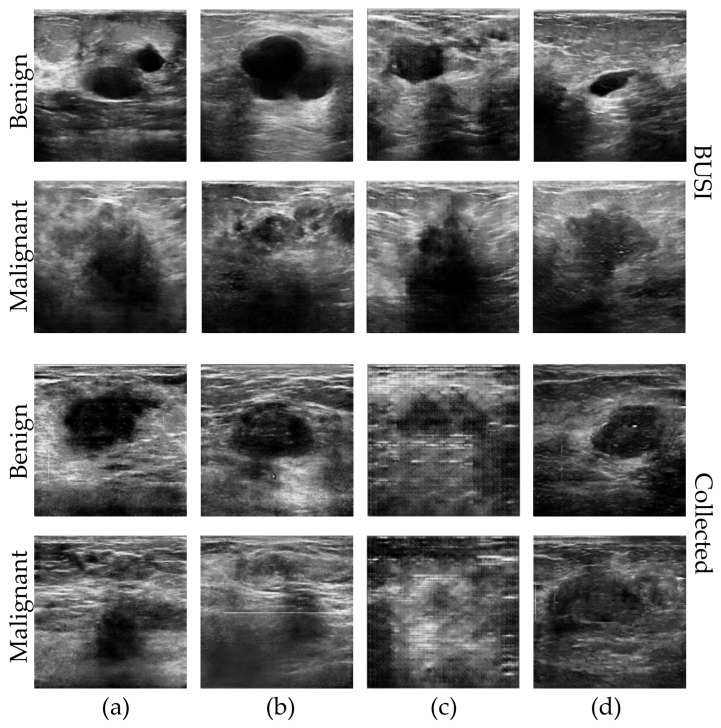
Examples of BUS image synthesis from generated masks for two datasets. From left to right, they are the results of different frequently used GAN methods and our proposed method: (**a**) images generated by DCGAN, (**b**) images generated by LSGAN, (**c**) images generated by WGAN, and (**d**) images generated by 2s-BUSGAN.

**Figure 10 sensors-23-08614-f010:**
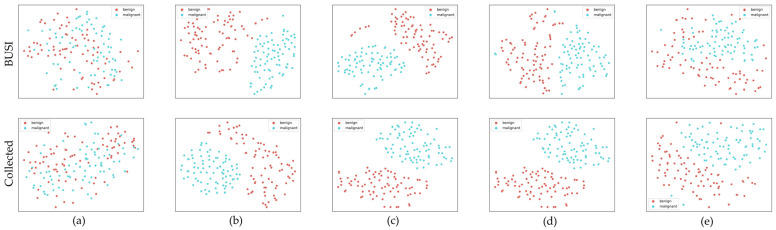
T-SNE embedding of benign (red) and malignant (blue) BUS images. From left to right, they are (**a**) GT images and the results of different GAN methods and our proposed method, (**b**) DCGAN, (**c**) LSGAN, (**d**) WGAN, and (**e**) 2s-BUSGAN.

**Figure 11 sensors-23-08614-f011:**
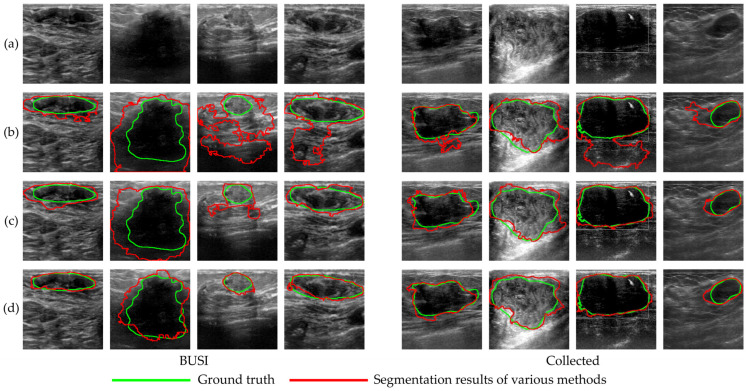
Results of breast tumor segmentation using different augmentation methods. Each column displays (**a**) the ground truth (GT) images and the predicted contours generated by the (**b**) baseline, (**c**) TA and (**d**) 2s-BUSGAN methods, respectively. The red curves represent the contours produced by each method, and the green curves indicate the contours marked by physicians.

**Table 1 sensors-23-08614-t001:** Quantitative results for IGS image synthesis. The downward arrows indicate that a lower value has better performance.

Dataset	Setting	Benign			Malignant		
		FID ↓	KID × 100 ↓	MS-SSIM ↓	FID ↓	KID × 100 ↓	MS-SSIM ↓
Collected	IGS w/o FML	167.1353	158.602	0.222	419.3311	580.4763	0.223
	IGS w/o DAM	142.3543	116.565	0.1422	188.7287	187.7055	0.1307
	IGS	89.1111	43.5631	0.1286	93.6413	48.6419	0.1285
BUSI	IGS w/o FML	208.5485	203.7132	0.2066	189.5309	184.7761	0.2618
	IGS w/o DAM	153.3376	130.2609	0.177	178.1268	179.8315	0.2843
	IGS	72.0245	27.7052	0.1509	101.0019	62.3815	0.2089

**Table 2 sensors-23-08614-t002:** Quantitative results for 2s-BUSGAN image synthesis. The downward arrows indicate that a lower value has better performance.

Dataset	Setting	Benign			Malignant		
		FID ↓	KID × 100 ↓	MS-SSIM ↓	FID ↓	KID × 100 ↓	MS-SSIM ↓
Collected	DCGAN	101.3506	60.6894	0.209	96.2566	63.4413	0.1736
	WGAN	186.4802	176.8394	0.3502	160.9212	144.1009	0.2556
	LSGAN	76.8730	41.055	0.2201	70.7381	33.7014	0.1908
	2s-BUSGAN	89.1111	43.5631	0.1422	93.6413	48.6419	0.1307
BUSI	DCGAN	62.2625	32.8401	0.1957	128.2199	108.9818	0.3507
	WGAN	120.3191	109.0968	0.1727	135.7969	119.5491	0.3150
	LSGAN	72.7607	38.9978	0.2074	92.5337	57.3108	0.3028
	2s-BUSGAN	72.0245	27.7052	0.1509	101.0019	62.3815	0.2089

**Table 3 sensors-23-08614-t003:** The proportion of which images are determined to be real of three doctors’ three types of images: the GT and the synthesized images by two different methods.

	Collected			BUSI		
	GT	LSGAN	2s-BUSGAN	GT	LSGAN	2s-BUSGAN
Doctor 1	0.75	0.60	0.57	0.85	0.47	0.47
Doctor 2	0.80	0.67	0.63	0.88	0.47	0.37
Doctor 3	0.70	0.67	0.60	0.83	0.53	043
All doctors	0.75	0.65	0.60	0.85	0.49	0.42

**Table 4 sensors-23-08614-t004:** The classification results of the benign and malignant of the GT and the synthesized images by two different methods by tree doctors.

	Collected			BUSI		
	GT	LSGAN	2s-BUSGAN	GT	LSGAN	2s-BUSGAN
Doctor 1	0.70	0.53	0.57	0.83	1.00	0.83
Doctor 2	0.63	0.57	0.63	0.82	0.93	0.73
Doctor 3	0.65	0.50	0.63	0.85	0.90	0.8
All doctors	0.66	0.53	0.61	0.83	0.92	0.79

**Table 5 sensors-23-08614-t005:** Evaluation of the classification performance with different data augmentation strategies.

Dataset	Setting	Precision	Recall	f1-Score	Accuracy
Collected	Baseline	0.63 ± 0.01	0.627 ± 0.006	0.623 ± 0.006	0.624 ± 0.007
	TA	0.64 ± 0.01	0.637 ± 0.012	0.637 ± 0.012	0.636 ± 0.010
	TA + LSGAN	0.66 ± 0.00	0.657 ± 0.006	0.667 ± 0.006	0.661 ± 0.005
	TA + 2s-BUSGAN	0.69 ± 0.026	0.69 ± 0.026	0.687 ± 0.021	0.688 ± 0.025
BUSI	Baseline	0.82 ± 0.017	0.817 ± 0.021	0.813 ± 0.023	0.816 ± 0.019
	TA	0.833 ± 0.012	0.833 ± 0.012	0.833 ± 0.012	0.836 ± 0.013
	TA + LSGAN	0.84 ± 0.00	0.823 ± 0.001	0.823 ± 0.006	0.821 ± 0.009
	TA + 2s-BUSGAN	0.857 ± 0.025	0.847 ± 0.015	0.847 ± 0.015	0.846 ± 0.015

**Table 6 sensors-23-08614-t006:** Comparison of the segmentation performance (DICE) with different data augmentation strategies.

Setting	BUSI		Collected	
	Benign	Malignant	Benign	Malignant
Baseline	0.7332	0.7354	0.6468	0.7021
TA	0.7422	0.7395	0.6588	0.7215
2s-BUSGAN + TA	0.7591	0.7565	0.6786	0.7316

## Data Availability

The data presented in this study are available on request from the corresponding author. The data are not publicly available due to ethical restrictions.

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
