# Peer review of "2S-BUSGAN: A Novel Generative Adversarial Network for Realistic Breast Ultrasound Image with Corresponding Tumor Contour Based on Small Datasets"

_sensors, 2023, doi:10.3390/s23208614_

Round 1

Reviewer 2 Report

The paper is well written, however, I would like the authors to have an explanation for these comments:

1. The methodology mentions training on imbalanced and small datasets. Given that the performance and generalization capability of a model are typically influenced by the quality and quantity of the training data:

  • How does the 2s-BUSGAN model ensure that the generated images are not merely reproductions of the training samples, given the small dataset scenario?
  • Could the authors provide any insights or results pertaining to the model’s generalization capabilities when applied to unseen, diverse, real-world datasets?

2. Given that IGS aims to incorporate texture details into the structurally informed images from MGS and FML is used to enhance image synthesis quality:

  • Could the authors elaborate on how they quantitatively and qualitatively evaluated the texture details and synthesis quality of the generated images?
  • How does the Feature Matching Loss (FML) contribute to the perceptual quality and clinical relevance of the generated images, and how is it validated against real-world clinical cases to ensure the synthesized details are not misleading or incorrect?

3. The methodology mentions using Spectral Normalization and other measures to stabilize the training of GAN. However, it would be helpful to have more detailed insights or discussions on the challenges faced during training, and any additional measures, tweaks, or optimizations performed to ensure stable and successful training of the model.

4. How were the values for the weight λ in the feature-matching loss term and the number of residual blocks in G2 determined? Were these values optimized based on preliminary experiments?

5. Could you elaborate more on the decision to resize all images in both datasets to 256×256 pixels? How does this resizing affect the model’s learning and performance, especially given the original average size of 500×500 pixels in the BUSI dataset?

6. Were the models in the study subjected to any form of hyperparameter tuning or optimization? If so, how were the optimal hyperparameters determined?

Could be improved

Round 2

Reviewer 2 Report

The authors have addressed all the comments.

The authors have addressed all the comments.